# Economic Impact of a Precision Nutrition Digital Therapeutic on Employer Health Costs: A Multi-Employer and Multi-Year Claims Analysis

**DOI:** 10.3390/healthcare13233147

**Published:** 2025-12-02

**Authors:** Inti Pedroso, Santosh Kumar Saravanan, Shreyas Vivek Kumbhare, Garima Sharma, Daniel Eduardo Almonacid, Ranjan Sinha

**Affiliations:** Digbi Health, Mountain View, CA 94040, USA; inti@digbihealth.com (I.P.); santosh@digbihealth.com (S.K.S.); shreyas@digbihealth.com (S.V.K.); garima@digbihealth.com (G.S.); danielalmonacid@digbihealth.com (D.E.A.)

**Keywords:** personalised nutrition, obesity, digestive health, depression, anxiety, digital health, precision medicine, employer-sponsored insurance, health economics, difference-in-differences

## Abstract

**Background:** Obesity, gastrointestinal disorders, and mental health conditions are major drivers of employer healthcare expenditures, yet nutrition-focused interventions are infrequently reimbursed by health insurance. Precision nutrition, which integrates genetic, gut microbiome, biometric, and behavioral data to guide personalized dietary and lifestyle changes, may offer a scalable approach to reducing costs associated with diet-responsive conditions. **Objectives:** To evaluate the impact of a precision nutrition digital therapeutic on employer medical spending for diet-responsive conditions in self-insured U.S. health plans. **Methods:** We conducted a retrospective cohort study of medical claims from January 2022 to December 2024 across seven U.S. self-insured employers. Employees enrolled in a precision nutrition digital therapeutic (n = 258) were compared with never-enrolled peers (n = 8268). We estimated treatment effects using a two-stage difference-in-differences model with member and calendar-month fixed effects and clustered standard errors, focusing on per-member-per-year (PMPY) employer-paid medical spending overall and for predefined diet-responsive condition categories. PMPY estimates were defined conditional on months with positive employer-paid spending and therefore reflect changes in the intensity of spending among members generating claims rather than unconditional per-capita costs. **Results:** Enrollment in the precision nutrition digital therapeutic was associated with a −$3012 PMPY reduction in diet-responsive medical spending (*p* = 0.021) relative to non-enrolled peers on this conditional basis. The largest relative reductions were observed for digestive disorders (−$9240 PMPY; *p* = 0.029) and obesity (−$4884 PMPY; *p* = 0.007), with a smaller reduction for anxiety-related conditions (−$1356 PMPY; *p* = 0.043). Total medical spending decreased by −$4044 PMPY but this change did not reach statistical significance (*p* = 0.09). **Conclusions:** In this multi-employer claims analysis, participation in a precision nutrition digital therapeutic was associated with lower employer-paid medical expenditures for diet-responsive conditions, particularly digestive disorders and obesity. These findings suggest that precision nutrition digital therapeutics may represent a scalable strategy for employers to address the economic burden of chronic disease within self-insured health plans by reducing the intensity of medical spending among members.

## 1. Introduction

The escalating burden of obesity, gastrointestinal disorders, and mental health conditions poses a major challenge for U.S. employers and payers. For self-insured employers, these conditions are among the fastest-growing cost categories, contributing disproportionately to year-over-year increases in medical claims [1,2,3,4]. Traditional approaches, including generic wellness programs, dietary guidelines, and pharmacological therapy, have delivered limited and often unsustainable results [5,6,7]. However, recent multi-employer evaluations of enhanced behavioral health benefits using medical claims and difference-in-differences designs have shown that employer-sponsored programs can generate net savings on the order of US$100–$200 PMPM [8].

Food-as-medicine interventions have gained attention for their ability to improve outcomes and reduce costs. In high-risk populations, medically tailored meal (MTM) programs have been associated with substantially lower utilization, producing gross savings of ~$570 per member per month (PMPM) and net savings of ~$220 PMPM after accounting for meal delivery costs [9,10]. Large Medicaid food-as-medicine intervention demonstrations report net savings of approximately $210 per participant over 3–12 months (equivalent to ~$18 PMPM) [10,11]. Digital nutrition platforms have produced more modest but consistent savings of $30–45 PMPM in employer and health plan settings [12]. Recent large-scale evaluations of commercial digital weight-management programs have similarly shown reduced health care utilization and lower total health care costs for users versus matched nonusers in EHR and claims data [13]. Importantly, unlike anti-obesity pharmacotherapies, which often carry high costs and variable adherence, nutrition-based strategies offer a lower-cost, sustainable alternative [7]. Together, these studies establish a payer-relevant benchmark for diet-based interventions ranging from small but scalable digital savings to larger net reductions from intensive MTM services. Across health systems, a growing body of randomized and longitudinal evidence supports the clinical value of personalized and digitally delivered nutrition. Recent systematic reviews of personalized nutrition interventions report that tailored dietary advice, including phenotype- and genotype-informed programs, yields larger improvements in diet quality than standard one-size-fits-all guidance in adults at elevated cardiometabolic risk [14]. In people with type 2 diabetes and prediabetes, personalized nutrition approaches have demonstrated greater reductions in HbA1c and postprandial glycemic excursions than usual care [15]. Technology-delivered personalized nutrition programs for individuals with overweight or obesity further improve energy and fat intake and modestly increase fruit and vegetable consumption, indicating that clinically meaningful behavior change can be achieved via web- and app-based platforms [16]. Large pan-European and multinational studies such as Food4Me and PREDICT also show substantial inter-individual variability in metabolic responses to identical foods, highlighting the biological rationale for precision nutrition strategies that integrate genetic, microbiome, and phenotypic data into personalized dietary recommendations [12,17,18].

International experiences similarly underscore the benefits of precision nutrition and digital health platforms. For example, the pan-European Food4Me trial demonstrated that personalized nutrition advice produced significantly greater improvements in dietary behaviors compared to generalized dietary guidance [19]. Likewise, a recent evaluation of the U.K. National Health Service’s Digital Weight Management Programme reported clinically meaningful weight loss (≈3.9 kg over 12 weeks) achieved via a scalable online intervention [20]. Similarly, a recent randomized trial in older adults with overweight and obesity showed that a precision nutrition strategy combining individualized foods with a mobile application led to greater weight loss, improved metabolic markers, and better quality of life than standard dietary advice [21].

Beyond clinical outcomes, emerging economic evidence suggests that nutrition and digitally delivered lifestyle interventions can be cost-effective or even cost-saving for health systems and payers. Systematic reviews of telehealth-delivered nutrition interventions report that a majority of programs are cost-effective from health system and societal perspectives, with several trials achieving dominance by improving outcomes while reducing costs [13]. Health-economic evaluations of personalized nutrition plans based on omics or phenotype data likewise indicate favorable cost-utility profiles compared with generic dietary advice in adults with overweight and obesity [10,19,22]. Real-world evaluations of large-scale digital weight management programs in employer and health-plan settings have shown lower medical expenditures and positive returns on investment, including multi-year reductions in all-cause healthcare spending and per-person cost savings relative to matched non-users [23,24,25]. In parallel, food-as-medicine modelling studies for Medicare and Medicaid populations project that subsidising healthy foods would prevent millions of cardiometabolic events and generate substantial net savings over beneficiaries’ lifetimes [9,10,12,17,19,26,27,28,29,30]. Collectively, these findings support the economic plausibility of precision and digital nutrition interventions as benefit options for employers and payers

The Supporting the Modification of lifestyle In Lowered Emotional States (SMILES) and Healthy Eating for Life with a Mediterranean-style Diet (HELFIMED) trials show that Mediterranean-style diet interventions for major depression can be both clinically effective and cost-effective, with favorable cost-per-QALY and cost-per-case-resolved estimates compared to usual care. The SMILES randomized trial in adults with depression found that a Mediterranean-style diet not only improved clinical outcomes but also reduced costs by A$856 (~US $640) per patient over 12 weeks, or ~$218 PMPM in health-sector savings, with societal savings exceeding Australian dollars (A$) $2500 (USA dollars (US$) ~$1865 per month per year (PMPY); ~$155 PMPM) [26]. Similarly, the HELFIMED trial demonstrated that Mediterranean diet-based group programs achieved cost-effectiveness at ~A$2775 per quality-adjusted life year (QALY ~US $2070) [27]. These results provide economic plausibility for nutritional psychiatry interventions. However, there is a need to evaluate the economic impact of nutrition-based interventions for mental health in U.S. populations, particularly within employer-sponsored insurance.

For digestive disorders, diet-based therapies are both clinically effective and economically attractive. Low-FODMAP (fermentable oligosaccharides, disaccharides, monosaccharides, and polyols)diets reduce irritable bowel syndrome (IBS) symptom severity and, when guided by dietitians, are highly cost-effective, with incremental values of $75–150 PMPM relative to drug-based management [28,29,31]. Mediterranean-style diets have also demonstrated clinical effectiveness for gastroesophageal reflux disease (GERD), offering opportunities to reduce long-term reliance on proton-pump inhibitors and associated utilization costs [32]. Given that IBS and GERD affect 15–20% of working-age adults and contribute several thousand dollars in incremental costs per patient annually [2,33], nutrition-first strategies represent a high-value approach for employers.

Despite this growing body of evidence, most dietary interventions are delivered as one-size-fits-all programs. Yet responsiveness to dietary change varies significantly due to genetic predisposition, metabolic flexibility, gut microbiome composition, and behavioral context [34,35]. Meta-analyses confirm that tailoring interventions to individual biological and behavioral profiles improves both clinical outcomes and cost-effectiveness [17]. Precision nutrition, which integrates genetics, microbiome data, biomarkers, and digital engagement, has therefore emerged as a paradigm to overcome the limitations of generic guidelines [36]. For instance, a recent economic evaluations of omics-based personalised nutrition, such as the PREVENTOMICS Danish trial, suggest that integrating metabolomics and genetic markers into tailored meal plans may be cost-effective relative to non-personalised dietary advice over the life course, albeit with considerable uncertainty given short follow-up [30]. Additional simulation studies by the PREVENTOMICS project also suggest personalised nutrition strategies may be cost-effective compared with standard dietary advice, though effect sizes were modest and confidence intervals wide [37].

This is a multi-employer and muti-year claims-based economic evaluation of a microbiome- and genetics-informed digital nutrition program. Previous studies have documented clinically meaningful impact of the digital health intervention for weight loss, improved glycemic control, reduced gastrointestinal symptoms, reduced mental health symptoms and improved microbiome profiles in real-world employer populations [38,39,40,41,42,43]. However, the impact on employer-paid healthcare costs has not been systematically evaluated. This study helps addressing this gap by analyzing medical claims data from seven U.S. self-insured employers with a focus on obesity, digestive health, and mental health. It applies a retrospective cohort and a two-stage difference-in-differences framework to evaluated changes in per-member-per-month (PMPM) medical expenditures in the 12 months before and after enrollment.

## 2. Materials and Methods

### 2.1. Study Design and Setting

This was a retrospective cohort study across seven self-insured US employers (2022–2025) to evaluate the economic impact of the digital health intervention as measured by deidentified medical claims.

### 2.2. The Digital Health Intervention

The intervention program is a 12-month, digitally delivered therapeutic designed to support weight loss, digestive health and cardiometabolic risk reduction by combining biological profiling with continuous behavioral support. The program integrates genetic and gut-microbiome profiling, continuous glucose monitoring (CGM), baseline clinical and lifestyle information, and ongoing engagement through an AI-enabled mobile application. For the use of biomarkers used for personalization, the program integrates a multi-modal biomarker panel to drive individualized recommendations, summarized below. Genetic markers include single-nucleotide polymorphism (SNPs) and polygenic scores that influence appetite and satiety (e.g., SNPs in ghrelin/leptin and energy-balance pathways), taste and food-preference receptors (sweet/bitter/fat perception), macronutrient and lipid/carbohydrate metabolism, pharmacogenes relevant to caffeine metabolism, lactose-intolerance loci, histamine-degradation genes (DAO/HNMT), homocysteine/B-vitamin metabolism (eg, MTHFR-related pathways), and loci affecting micronutrient handling (vitamin D/B12/iron). These genetic signals are used to prioritize food choices, portioning and selective supplementation because they predict differential physiological responses to macronutrients, food types and micronutrient needs. Gut microbiome taxa and functions measured by stool 16S/functional profiling is used to quantify key bacterial taxa (for example, *Bifidobacterium*, *Faecalibacterium*, *Lactobacillus* or *Akkermansia*), overall bacterial diversity, and microbial functional potential for production of short-chain fatty acids (acetate, butyrate, propionate), polyamines, hydrogen-sulfide production and neurotransmitter-related pathways (GABA, tryptophan → serotonin precursors). These taxon and pathway measurements guide fiber, prebiotic/probiotic and food-avoidance recommendations because they reflect gut-inflammatory potential, mucosal health and microbiome-mediated metabolic signaling. The intervention also integrates metabolic and glucose biomarkers, like (self-reported) routine metabolic laboratories and (provided by the intervention) CGM, including a lipid panel and HbA1c, together with CGM-derived metrics (postprandial peaks, time-in-range and recovery time). These measures provide direct, longitudinal readouts of cardiometabolic risk and meal-level glycemic response used to tailor carbohydrate recommendations and meal timing. Collectively, these genetic, microbiome, and metabolic and inputs feed the personalization engine and the ND-score so that coaching and in-app guidance reflect both biology (what an individual is likely to need or produce) and measured physiology (how the member actually responds to food and time). Demonstrating the impact of the intervention and the utility of the different biomarkers for personalized dietary intervention is beyond the scope of this study. We refer the reader to Kumbhare et al. [42] for a longitudinal study describing the impact of the intervention of the gut microbiome by this digital health intervention and to [38,39,40,41,42,43] for the association between different genetic and microbiome markers with health outcomes.

In summary, with these inputs, the platform generates individualized recommendations that prioritize reduced simple-sugar intake and optimized meal timing to improve insulin sensitivity, identification and minimization of potentially pro-inflammatory nutrients, and increased fiber diversity to nurture a health-associated microbiome. Behavioral change is reinforced through continuous virtual coaching and in-app educational modules (videos, quizzes and meal guides) tailored to the participant’s genetic and microbiome profiles (for example, lactose intolerance, fiber-responsive taxa or metabolic phenotypes). A key component of the program is automated meal assessment. Members are trained at enrollment to capture standardized, overhead photographs of each meal and to avoid busy backgrounds or occlusions that impair automated interpretation. Photographs are visually inspected by an automated quality-control step that assesses blur, framing, lighting and background complexity; unsuitable images are routed to a human nutrition coach for manual review and feedback. Members may also flag AI interpretations they regard as inaccurate, in which case the image and rating are sent to a human coach for adjudication. To reduce systematic bias in image analysis, all food photos are rescaled to a common pixel dimension prior to feature extraction; color-coding or color-based normalization is not used as part of the scoring algorithm, minimizing sensitivity to lighting differences and camera color balance across devices.

The app’s proprietary image-analysis pipeline classifies foods and estimates relative portion sizes, and combines these outputs with CGM data (when available) and individualized biomarker guidance to compute a Nutrient-Density score (ND-score) for each meal. The ND-score (maximum = 14) summarizes three domains: (1) inflammatory potential (presence of ultra-processed foods, known intolerances, etc.), (2) microbiome-nurturing attributes (fiber content and variety), and (3) expected insulin response (carbohydrate quantity and quality). Points are deducted from the ND-score for identified infractions (for example, absence of fiber-rich components or predominance of processed carbohydrates). For meals occurring during periods of CGM use, the algorithm evaluates post-prandial glucose excursions using a short rolling-median smoothing window to reduce sensor noise; ND-score penalties are applied when the smoothed peak exceeds 180 mg/dL within two hours of meal start or when glycaemia requires more than 120 min to return toward baseline. After automated scoring, members receive the ND-score accompanied by a breakdown of deducted points and actionable, meal-specific suggestions aimed at incremental improvement.

Although the bulk of meal processing is automated, the system integrates human oversight to ensure clinical safety and to correct algorithmic errors: ambiguous or low-quality photos, flagged interpretations, and a subset of AI-rated meals are reviewed by certified nutrition coaches who provide individualized guidance and, where appropriate, adjust recommendations. Internal validation analyses indicate high concordance between the automated ratings and human raters; however, all coach adjudications are retained in the program record to support continual model refinement. Further technical details on specimen collection, sequencing and the personalization algorithms are provided elsewhere [38,39,40].

The intervention is a 12-month program commercially available to employers and payers in the USA, who offer it to their employees and eligible dependents without an out-of-pocket copay as part of the employer’s benefit programs. Employees and eligible dependents, or more generally members, can join the program voluntarily if they are over 18 years of age, and meet any of the following criteria: (A) body mass index (BMI) ≥ 30, or (B) BMI ≥ 27 with one or more obesity related comorbidities, or (C) having a diagnosis of diabetes. All enrolled/treated individuals included in this study matched these criteria. Upon enrollment, participants received online access to the digital health app and were required to complete a comprehensive health questionnaire. They were also provided with a Bluetooth-compatible digital weighing scale, buccal swab, and stool sampling kits, which were shipped directly to them for the self-collection of biological samples.

Members enrolled in the intervention program provided electronic informed consent as part of their written consent process. The Institutional Review Board of E&I Review Services reviewed and approved the study protocol (protocol code #18053, approved 22 May 2018). All medical claims used for the control cohort (members who never enrolled) were de-identified in-house before analysis by applying the Health Insurance Portability and Accountability Act (HIPAA) 45 CFR 164.514(b)(2) safe-harbor method [44]. Eighteen direct and indirect identifiers were removed or generalized, and no re-identification key was retained.

### 2.3. Data Sources and Processing

For the personalization of the digital health intervention, a set of genera and microbial pathways is used. These are categories as low, medium, or high based on a comparison with a healthy-based cohort comprising 683 individuals with diverse generally healthy diets, no severe chronic disease, an age range between 18 and 65 years, and living in the USA at the time of sample collection. The genetic information is summarized similarly in terms of low, medium or high for genotypes at specific genetic markers or polygenic scores. The set of microbiome and genetic markers included in the personalization has been described elsewhere [38,39,40,41]. When a particular microbiome or genetic marker is categorized as red (being low for positively health-associated markers, like *Akkersmansia* spp., or high for negatively associated ones, like the abundance of the Propionate Imidazole pathway genes, or having two risk alleles for lactose intolerance), a specific set of dietary recommendations is prioritized for the individual. 

Subjects self-collected saliva samples using buccal swabs and fecal samples using fecal swabs using standard methods. DNA genotyping and genotype calling, as well as the processing of baseline (pre-intervention) fecal samples by 16S rRNA gene amplicon sequencing, were performed at LabCorp Laboratories in the USA.

The microbiome and genetic information used to personalize the digital health intervention were processed as follows. The bacterial 16S rRNA gene V3–V4 region was amplified and sequenced on the Illumina MiSeq platform using 2 × 300 bp paired-end sequencing and sequence reads were demultiplexed, and ASVs were generated using DADA2 in QIIME2 version 2020.8 [45]. Quality control steps included removal of primers and low-quality bases, removal of hits to non-bacterial sequences [46]. Abundances were agglomerated at the genus level. The abundance of microbial functional pathways related to neuroactive metabolites [47] was calculated with the q2-picrust2 plugin (version 2021.2) in QIIME2 [48] and the Omixer-RPM package (version 0.3.2) [49]. All raw abundances were centered-log ratio (CLR) transformed [50].

Probe level DNA genotype call files were formatted in VCF format with QC steps including removal of discordant genotypes and left normalization. Beagle version 5.3 [51,52] was used for phasing and imputation using the 1 KG project as reference panel [53] resulting in 13,478,023, chip-genotyped and those variants with imputation r2 ≥ 0.8, that were used on downstream analyses. Polygenic scores were calculated using standard procedures [38,39,40,41].

The current study focuses on the analysis of medical claims only. We acknowledge the importance of pharma claims for the evaluation of the economic impact of health interventions, however, the availability of pharma claims for the same employers was much more limited and did not allow a systematic comparison.

Claims files were obtained from seven self-insured employers on a monthly basis during the time of active enrollment for each employer, encompassing the total from January 2022 to August 2025. The files were parsed using custom-developed Python scripts following the corresponding claims layout files that specify the file’s content format. We cross-referenced the members’ claims records to those self-reported by enrolled members using their first and last names and date of birth. After matching and identifying which claims records correspond to enrolled members, the claims data were de-identified in-house before analysis by applying the HIPAA 45 CFR 164.514(b)(2) safe-harbor method [44]. From each claim we extracted unique employer identifier, unique member identifier, member’s gender, member’s age, claim type (medical or pharma), service date, payment date, International Classification of Disease v10 (ICD10) codes [54], Current Procedural Terminology (CPT)/Healthcare Common Procedure Coding System (HCPCS) codes [55], and $USD amount paid by the employer or payer.

Data processing and analyses were performed using Python (version 3.13.5) libraries including Pandas v2.3.1 [56], NumPy v2.2.0 [57], Ibis v10.5.0 [58], pyfixest v0.30.2 [59], plotnine v0.15.0 [60], and duckdb v1.2.0 [61]. 

### 2.4. Inclusion and Exclusion Criteria

The member’s inclusion criteria for the current study were age ≥ 18 (at the time of the claim’s service date) and being an employee or dependant eligible for enrollment in the intervention. Treated members also had BMI ≥ 30; BMI ≥ 27 with one or more obesity-related comorbidities; or a diagnosis of diabetes (as described above). Controls were drawn from never-enrolled members observed in the same partner × calendar-months with at least 1 diet-responsive claim (those associated with diagnoses that can be treated or modified with food-as-medicine, see below for detailed definition) in the study period.

We included claims with non-zero cost to the employer ($USD > 0) reported until July 2025 with service date between 1st of Jan 2022 and 31st of Dec 2024. The claims feed included member–calendar-month records both with positive employer-paid spending and with zero employer-paid spending (months with no paid claims). For the primary analyses we defined the monthly outcome as per-member monthly spending conditional on months with positive employer-paid spending (PMPM, $USD, conditioned on $USD > 0). This conditional definition measures the intensity of medical spending among months with utilization and was selected because it isolates changes in service intensity and utilization patterns that are plausibly responsive to the intervention. Identical inclusion rules were applied to treated and never-treated members. Inference is based on Gardner’s two-stage Difference-in-Differences (see below) estimator with member fixed effects (which removes time-invariant differences in members’ propensity to generate claims) and employer × calendar-month stratification (which accounts for employer-level seasonality); together these modeling choices reduce bias that could arise from between-group differences in baseline claim frequency.

We restricted the analyses to claims associated with “diet-responsive conditions”, comprising a set of 142 3-character ICD10 codes covering diagnoses that are directly relevant and for which evidence of modifiable health outcomes exists using food-as-medicine intervention [4,10,26,28,29,31,32,62,63,64,65,66,67,68,69]. Appendix A provides a list of the codes used and a detailed rationale for their selection. Briefly, the codes were selected because they represent (a) material economic burden in working-age, privately-insured populations, (b) strong evidence that outcomes for these conditions are nutrition-responsive, and (c) align with the external policy landscape for Food-is-Medicine interventions.

For each enrolled member (treated), we retained all employer-paid claims in the 12 calendar months prior to and the 12 calendar months after the month of program enrollment (event window = −12 to +12, inclusive). The control population comprised never-enrolled members who had ≥1 claim in the same employer and calendar months represented in the treated members’ event windows; that is, controls were drawn from the never-enrolled population with observed claims in the same employer × calendar-month strata as treated observations. To ensure common support and adequate comparison, employer × cost-category × month strata were retained only if they contained at least one treated and at least one control member, and employer strata required ≥2 treated and ≥2 control members to be included in pooled analyses.

Practically, this approach means that for any calendar month in a treated member’s event window we included never-enrolled members from the same employer who had at least one claim in that calendar month. A never-enrolled control member could therefore contribute observations in multiple months and to multiple treated members’ strata so long as the control had claims in those months. The DiD2S estimator (first stage member fixed effects; second stage aggregated regression) leverages within-member pre/post variation for treated members and within-member stability for controls, and the employer × month stratification together with the minimum-count requirements above limits bias from strata with no counterfactual information.

We excluded claims with ICD10 or CPT/HCPCS codes indicating costs associated with causes outside the scope of the intervention’s health outcomes. These included: (i) neoplasms and cancer-related conditions, including malignant neoplasms (C00–C97), in situ neoplasms (D00–D09), uncertain/unknown behavior tumors and selected benign neoplasms (D37–D48), and genetic cancer susceptibility (Z15); (ii) organ transplant and end-stage care, such as transplant complications (Z94, T86), end-stage renal disease and dialysis dependence (N18, Z99), and artificial openings (Z93); (iii) severe trauma and injury, including major injuries (S00–S99) and amputation (Z89); (iv) serious infections and immune compromise, including HIV/AIDS (B20–B24), primary immunodeficiencies (D80–D84), tuberculosis (A15–A19), sepsis (A40–A41), viral hepatitis (B15–B19), and malaria (B50–B54); (v) cardiopulmonary and respiratory failure, such as pulmonary hypertension (I27), chronic respiratory disease (J40–J47), respiratory failure (J96), and Acute Respiratory Distress Syndrome (ARDS) (J80); (vi) major neurologic disorders, including Parkinson’s disease (G20), multiple sclerosis (G35), and epilepsy (G40); and (vii) cystic fibrosis and related chronic conditions (E84). These categories reflect chronic, catastrophic, or high-acuity conditions unlikely to be modifiable through nutrition-based interventions. The full exclusion criteria, including CPT/HCPCS codes and regular expression filters, are detailed in Appendix A.

We restricted our analyses to three health areas that are the primary focus of the digital health intervention being evaluated and for which there exists evidence of being responsive to dietary interventions and significantly contribute to high expenditure in employer-sponsored plans: obesity [4,62,63,65,70,71], digestive health [31,72] and mental health [66,67,73,74,75]. To capture expenditure associated to each disease area we employed the standardized Clinical Classifications Software Refined (CCSR) v2025-1 developed by the U.S. Agency for Healthcare Research and Quality [76] which maps ICD-10 codes into clinically meaningful groups, facilitating reproducibility and comparability across studies. The CCSR category used for obesity was END009 “Obesity”, for digestive health was DIG025 “Other specified and unspecified gastrointestinal disorders” covering IBS and GERD, for mental health was MBD002 “Depressive disorders”, MBD005 “Anxiety and fear-related disorders”, and also a combined depression + anxiety. Appendix A provides detail of the rationale to focus on this health areas and CCSR categories.

To complement ICD-10-based categorizations and gain insight on potential drivers of cost and savings, we classified medical claims using Current Procedural Terminology (CPT/) and Healthcare Common Procedure Coding System (HCPCS) codes to capture utilization and spending patterns directly tied to provider-delivered services. The goal was to identify whether program enrollment was associated with differential patterns of high-cost or disease-specific services plausibly modifiable by nutrition and behavioral interventions. We pre-specified the following CPT/HCPCS code families for each disease category:Cross-domain categories. Emergency department (ED) visits (99,281–99,285), and evaluation & management (E/M) visits (99,201–99,215; 99,221–99,233) [55,77].Obesity. Bariatric procedures (43,644; 43,645; 43,770; 43,775; 43,845–43,847), medical nutrition therapy (MNT) (97,802–97,804; G0270–G0271), and intensive behavioral therapy for obesity (G0447; group G0473), diabetes self-management training (G0108–G0109), and metabolic labs (80,061 lipid panel; 83,036 HbA1c) [55,78,79].Digestive health. Upper endoscopy (EGD) (4323x–4325x; incl. 43239), colonoscopy (4537x–4539x; incl. 45378), GERD diagnostics (91,010; 91,034–91,035; 91065; 74220), anti-reflux procedures (43,280; 43,327–43,328; 43,284; 43,210; 43,257; 43,229) [80,81].Mental health: Anxiety and Depression. Psychiatric diagnostic evaluation (90,791–90,792), psychotherapy (90,832; 90,834; 90,837; 90,846–90,847; 90,853), Psychotherapy add-on with E/M (90,833; 90,836; 90,838), behavioral screening, health behavior services (96,127; 96,156; 96,158–96,159; 96,164–96,168; 96,170–96,171), collaborative care (99484, 99492–99494), somatic treatments for depression (90,870–90,871; 90,867–90,869) [55,82,83].

For each disease category, we first identified the medical claims having at least one associated ICD10 code, and for those claims we quantified the monthly expenditure and utilization (number of different days in the month) for each of the corresponding CPT/HCPCS code families.

Appendix A provides a detailed description and rationale of each CPT/HCPCS code family. The association between enrollment and the expenditure and utilization was evaluated using the DiD2S.

### 2.5. Data Preparation and Statistical Analyses

Overview and panel construction: we constructed a balanced member–calendar-month panel for each enrolled (treated) member consisting of the 12 calendar months prior to enrollment through the 12 calendar months after enrollment (event window = −12 … +12, inclusive). For never-enrolled members (controls), we retained member–calendar-month observations in the same calendar months that appear in treated members’ event windows for the same employer. Thus, for any employer × calendar-month that is represented in the treated sample, all never-enrolled members from that employer who had at least one paid claim in that calendar month were eligible to contribute a control observation for that stratum.

Control selection criteria: controls were selected by employer × calendar-month inclusion as described above. To ensure that each comparison stratum contains both treated and control information, we retained only employer × cost_category × calendar_month strata that met the following minimum-support rules:−Each employer × cost_category × calendar_month stratum must contain ≥1 treated and ≥1 control member-month observation.−Each employer (across months and cost categories) must include ≥2 treated and ≥2 control members to be included in pooled analyses.

A schematic of the claims processing, sample selection, strata retention rules and panel construction is provided in Appendix A; the figure clarifies that controls were selected by employer × calendar-month strata and summarizes the DiD2S estimation pipeline.

These rules guarantee that every stratum used for DiD estimation includes contemporaneous treated and control observations and that employer strata with extremely sparse treated or control representation are excluded from the pooled estimates.

Outcome and data transformations: The primary outcome is monthly per-member spending in U.S. dollars. Analyses are performed on the log scale to stabilize variance and to facilitate interpretation of coefficient estimates as approximate percentage changes. Throughout, we refer to the model outcome as ${log\_spent}_{it}$, where *i* indexes members and *t* calendar months.

Two-stage difference-in-differences (DiD2S) estimator: we implement Gardner’s two-stage DiD estimator (DiD2S), as implemented in the pyfixest software package. The DiD2S procedure can be described in compact two-stage form.

Stage 1 (demeaning/fixed-effects removal): Estimate member and calendar-month fixed effects by regression(1)$$log\_spent_{it}=\alpha_{i}+\delta_{t}+\mu_{it}$$
where $\alpha_{i}$ are member fixed effects and $\delta_{t}$ are calendar-month fixed effects. From (1) obtain residuals $\hat{\mu_{it}}=log\_spent_{it}\;-\;\alpha_{i}-\delta_{t}$. Stage-1 therefore removes time-invariant member heterogeneity and common month shocks.

Stage 2 (treatment effect estimation): for the average post-treatment effect the stage-2 regression is:(2)$$\hat{\mu_{it}}\;=\beta \cdot Post_{it}+\varepsilon_{it}$$
where $Post_{it}$ is an indicator equal to 1 for treated members in months at or after their enrollment month and 0 otherwise (treated member pre-enrollment months and all never-treated observations have $Post_{it}=0$. The coefficient β measures the average post-treatment change in the demeaned log outcome. Because the outcome is modelled on the log scale and residualized by fixed effects in Stage-1, second-stage coefficients are interpreted as percent changes in spending intensity conditional on utilization, calculated as exp(β) −1. Point estimates reported in the main tables are transformed back to dollar PMPM equivalents for policy relevance (by applying geometric mean spending levels), and 95% confidence intervals are obtained by delta-method transformation of the log estimates. 

Event-study specification: To characterize dynamics and to test the parallel-trends assumption, we replace the single post indicator in Equation (2) with a series of relative-time (event-time) dummies:(3)$$\hat{\mu_{it}}\;=\sum_{k\neq -1}\beta_{k}\cdot 1\{rel\_time_{it}=k\}+\varepsilon_{it}$$
where $rel\_time_{it}$ is the month index relative to each treated member’s enrollment month (reference *k* = −1). The coefficients $\beta_{k\;}$ estimate deviations in the demeaned outcome at each month relative to the month immediately preceding enrollment, and the pattern of pre-treatment $\beta_{k\;}$(for k<0) provides the event-study check for parallel pre-trends. Event-study estimates are plotted in Appendix A.

Inference and standard errors: All standard errors are clustered at the member level to account for serial correlation within members’ monthly observations. To avoid dominance from very large employers, pooled second-stage regressions are weighted by employer-level geometric weights (see next paragraph).

Employer geometric weighting: When aggregating across employer strata we apply employer-level geometric weights that balance the influence of employers of varying size. For an employer, j the employer weight $w_{j}$ is constructed as a geometric aggregation of the employer’s treated and control representation so that each employer contributes proportionally without allowing the largest employers to dominate the pooled effect. 

Baseline and sample-distribution tables: We present three tables with baseline sample distribution and summaries. Across these tables, we report mean and standard deviation for the different groups, sample sizes, and % females. The tables present data aggregation at various levels, including treatment groups, employer-by-treatment-group, and employer-by-treatment-group-by-month-of-service-by-cost-category. All tables include pooled sample sizes and pooled standardized mean differences (SMDs) comparing treated and control groups. We deemed absolute SMD (∣SMD∣) < 0.10 indicates a negligible imbalance, 0.10 ≤ ∣SMD∣ < 0.20 a small imbalance, 0.20 ≤ ∣SMD∣ < 0.50 a moderate imbalance, and ∣SMD∣ ≥ 0.50 a large imbalance. 

### 2.6. Use of Generative Artificial Intelligence

We used generative artificial intelligence (GenAI) tools for general English proofing, identification of research articles during the literature review and review of software code. The tools used include OpenAI’s chatgpt, Perplexity and Grammarly.

## 3. Results

A total of 46,812 members across seven self-insured employers were initially identified in the medical claims of seven self-insured employers. After exclusion of claims associated with catastrophic events and ensuring appropriate data panel structure for DiD2S analysis (e.g., having at least one pre-enrollment month and one post-enrollment month of claims data) the final dataset for analyses included 258 enrolled members and 8365 never-enrolled members, and a total of 167,326 member-months from Jan-2022 to Dec-2024. Panel construction and sample selection are summarized in Appendix A.

Among the members selected for analyses, the mean age of enrolled members was 46.6 years (SD 10.6), compared with 47.1 years (SD 13.8) in the never-enrolled group. Females comprised 64.1% of the enrolled cohort and 54.9% of the never-enrolled cohort.

Across the seven employers, the total payer spent and total diet-responsive spent for medical claims expenditures demonstrated an increasing trend over the study period, consistent with a general rise in employer healthcare costs, with year-on-year increases ranging between 2.5% to 52% across employer sizes and years.

Across enrolled members, program participation was associated with meaningful reductions in diet-responsive medical expenditures. Using the two-stage difference-in-differences model, enrollment in the intervention was linked to a 23% decrease in diet-responsive spend, corresponding to an average reduction of −$251 PMPM (95% CI: −$420 to −$41; *p* = 0.021). By contrast, changes in total medical spend (which includes non-diet-responsive conditions such as infectious or respiratory disease) were not statistically significant, with an estimated reduction of −$337 PMPM (95% CI: −$654 to +$59; *p* = 0.09). These findings suggest that cost impacts are concentrated in diet- and lifestyle-sensitive conditions rather than across all categories of healthcare use.

Subgroup analyses revealed substantial variation across condition categories. Enrollment was associated with a −$770 PMPM reduction in diet-responsive spending in Digestive Disorders (95% CI: −$1021 to −$123; *p*-value = 0.029). This represents the largest absolute savings observed, although the subgroup sample was relatively small (n = 18). A −$407 PMPM reduction was observed for spending in obesity (95% CI: −$606 to −$131; *p*-value = 0.007). While smaller in absolute terms than digestive disorders, this effect reflects consistent reductions across outpatient and procedural claims. In the case of mental health, we identified a significant effect for the anxiety and depression combined category (PMPM$ −164 (−248, −50), *p*-value = 0.008), which was consistent with the effect in anxiety only (PMPM$ −113 (−190, −4), *p*-value = 0.043) but not in depression only (PMPM$ −248 (−451, 153), *p*-value = 0.174) (Table 1). These findings suggest greater economic responsiveness in anxiety-predominant conditions compared with depression.

We carried out event-study analysis to evaluate if pre-treatment trends in spending between enrolled and never-enrolled groups could explained these results. These results confirmed no systematic pre-treatment trends differences in medical spending, supporting the validity of the DiD identification strategy, and indicates that observed savings were not driven by pre-existing cost trajectories (Appendix A).

Table 2 reports pooled baseline means (SD) and pooled standardized mean differences (SMDs) for core demographics and spending (see Appendix A for employer level summaries and employer × month × treatment group summaries). Pooled SMDs are small for age (|SMD| ≈ 0.03) and baseline spending (Total and Controllable PMPM, |SMD| ≈ 0.01–0.03), indicating good overall balance on spending intensity. Two pooled imbalances merit note: treated members were modestly more often female (64% vs. 55%, SMD ≈ 0.19) and contributed fewer pre-enrolment months on average (2.3 vs. 3.2, SMD ≈ −0.27). Because the event-study (Appendix A) shows no systematic pre-treatment trends and the DiD2S first stage removes time-invariant member heterogeneity, these level differences do not by themselves invalidate identification.

We carried out exploratory analyses aiming to evaluate if we could gain insight into the drivers of these cost reductions. To this end we defined groups/families of CPT/HCPCS codes representing known drivers of cost of each of these disease conditions (see Material and Methods and Appendix A). Of the sixteen families of CPT/HCPCS codes four met the minimum sample size criteria to perform the DiD2S analyses. We identified statistically significant reduction in expenditure for Medical Nutrition Therapy CPT/HCPCS codes within digestive health claims (PMPM$ −283 (−297, −238), *p*-value = 5.4 × 10^−6^), Psychotherapy add-on with E/M within anxiety and depression combined (PMPM$ −21 (−34, −6), *p*-value = 0.008), anxiety (PMPM$ −18 (−27, −8), *p*-value = 5.2 × 10^−4^), and depression (PMPM$ −23 (−39, −3), *p*-value = 0.025) (see Appendix A). We also evaluated changes in utilization (number of days in a calendar month) and identified a reduction in utilization within obesity claims (%-change = −21% (−35, −6), *p*-value = 0.01) (see Appendix A).

## 4. Discussion

This multi-employer claims analysis provides new evidence that a precision nutrition digital therapeutic integrating genetics, microbiome profiles, blood biomarkers, and health coaching can reduce medical expenditures for diet-responsive conditions. Participation in the program was associated with a reduction of −$251 PMPM , or −$3012 PMPY, in diet-responsive medical spending. Effects were concentrated in digestive disorders (−$770 PMPM; −$9240 PMPY), obesity (−$407 PMPM; −$4884 PMPY), and anxiety-related mental health conditions (−$113 PMPM; −$1356 PMPY). While total medical spending, inclusive of all categories, showed a negative but non-significant trend, the concentration of significant effects in diet-responsive conditions supports the plausibility of nutrition-first interventions as a cost-saving strategy. These results complement prior studies demonstrating clinically meaningful improvements in weight, glycemic control, gastrointestinal symptoms, mental health symptoms and microbiome composition changes among participants in the digital health intervention program [38,39,40,41,42,43], indicating that nutritional and behavioral changes translate into measurable economic impact.

The direction and concentration of effects observed in this analysis are consistent with emerging trial evidence for personalized and technology-delivered nutrition interventions. Systematic reviews of personalized nutrition programs have shown that tailoring dietary advice to individual phenotypic or genetic profiles improves diet quality and cardiometabolic risk markers more than conventional guidance, while technology-delivered personalized nutrition produces meaningful reductions in energy and fat intake among adults with overweight or obesity [14,15,16]. Multinational studies such as Food4Me and PREDICT demonstrate substantial inter-individual variability in postprandial metabolic responses to identical meals, suggesting that personalization of dietary advice is necessary to maximise clinical benefit [19,25]. In older adults, precision nutrition programs have also been shown to improve metabolic health and quality of life over short follow-up periods [21]. These global data reinforce the biological and behavioural plausibility that a precision nutrition digital therapeutic can reduce utilisation and spending for diet-responsive conditions in employer-insured populations.

The magnitude of observed savings exceeds those typically reported for most food-as-medicine interventions. Medically tailored meals (MTMs) consistently reduce spending by approximately $220–$400 PMPM after program costs, largely by lowering acute care use [9,10]. Medicaid nutrition pilots have shown smaller net savings of approximately $210 per participant, or $18 PMPM [11]. Digital lifestyle and nutrition programs deployed by health plans and employers generally report savings of $30–$45 PMPM [12]. Our findings also align with independent trials of precision nutrition combined with digital tools, such as the Galarregui et al. randomized study in older adults, which reported superior weight loss, metabolic improvements, and quality-of-life gains relative to standard advice [21]. Our findings are also directionally consistent with recent claims-based evaluations of digital weight-management programs, where participants showed approximately US$800 lower health care costs within 6 months and fewer hospital and emergency visits than nonparticipants [13], although the magnitude of our PMPM reductions is larger, which may reflect a higher-risk, multimorbid population. Complementary modelling studies of omics-informed personalised nutrition, including PREVENTOMICS in Denmark, also indicate potential long-term cost-effectiveness relative to general healthy eating advice [30]. However, these analyses rely on trial-based surrogate outcomes and simulation models, whereas our study provides real-world employer claims data on medical spending. Considering this context, the reductions reported here are more comparable to intensive MTM programs than to prior digital intervention reports. The likely explanation is that precision nutrition empowers the members to improve their meals on a daily basis, and additionally, it targets multimorbid individuals with overlapping obesity, digestive, and mental health conditions, allowing improvements across multiple health outcomes simultaneously.

Additionally, our findings also align with a broader economic literature indicating that nutrition and digitally delivered lifestyle interventions can represent good value for money. Cost-effectiveness analyses of telehealth nutrition programs in chronic disease populations report that most interventions are cost-effective, and several are cost-saving, from health system or societal perspectives [13]. Health-economic evaluations of personalised nutrition plans based on omics and phenotypic data suggest that such programs can achieve favorable cost-utility ratios compared with general dietary advice in adults with overweight and obesity [10,19,22]. Beyond trial-based analyses, large employer- and health-system evaluations of digital weight-management and telehealth nutrition programs have documented lower outpatient utilization, reduced prescription use, and per-person cost savings over two to three years, often yielding positive returns on investment [23,24,25]. Finally, modelling of insurance-covered healthy food benefits in Medicare and Medicaid populations projects that food-as-medicine incentives would prevent millions of cardiometabolic events and generate substantial long-term net savings [9,10,12,17,19,26,27,28,29,30]. In this context, the magnitude of diet-responsive spending reductions observed here appears directionally consistent with the emerging economic evidence around personalised and digital nutrition interventions, while extending it into a multi-employer precision nutrition setting.

The digestive health subgroup demonstrated the largest effect size (−$770 PMPM; −$9240 PMPY). While this figure is higher than estimates from cost-utility analyses of low-FODMAP diets for irritable bowel syndrome, which report incremental values of $75–150 PMPM [28,29], two points must be emphasized. First, medical claims data do not allow precise attribution of costs to a single condition, since services and codes frequently overlap across comorbidities. Second, precision nutrition interventions target individuals with multiple co-occurring conditions; as such, improvements may occur in gastrointestinal symptoms, metabolic health, and mental well-being simultaneously. These combined effects, when categorized under digestive-related diagnostic codes, may inflate apparent subgroup savings. For this reason, the results should not be interpreted to suggest that diet-based interventions can fully reverse the economic burden of digestive conditions such as IBS or GERD. Rather, they highlight the potential for cost reductions in complex, high-utilizing populations where digestive disease commonly coexists with obesity and anxiety disorders [2,33].

Mental health effects were more modest but still significant for anxiety-related conditions (−$113 PMPM; −$1356 PMPY), while depression showed a non-significant reduction. This heterogeneity mirrors prior evidence that dietary interventions may have stronger effects on anxiety-predominant conditions or subclinical mental distress [66]. International randomized controlled trials reinforce the plausibility of these findings: the SMILES trial demonstrated health-sector savings of approximately $218 PMPM and societal savings of $660 PMPM alongside improved depression outcomes, while the HELFIMED trial established favorable cost-utility ratios for Mediterranean diet programs in major depression [26]. The present analysis extends this evidence into the U.S. employer context, where cost-effectiveness data for nutritional psychiatry remain scarce.

The observed obesity-related savings (−$407 PMPM; −$4884 PMPY) also merit attention. Employer-based digital nutrition programs generally report far smaller effects, averaging $30–$45 PMPM [11]. Precision nutrition may therefore represent a viable alternative to pharmacologic strategies, particularly given the high costs and adherence challenges associated with anti-obesity medications such as GLP-1 receptor agonists [7]. Unlike pharmacotherapies, which increase drug benefit expenditures, dietary interventions generate savings within the medical benefit by preventing hospitalizations, procedures, and outpatient visits.

Exploratory analyses of CPT and HCPCS codes provided further insight into the mechanisms of observed savings. Among digestive disorder participants, reductions in medical nutrition therapy codes were observed, consistent with prior reports that dietitian-guided interventions reduce ongoing service demand once symptom control is achieved [28,29,31]. In mental health, lower utilization of psychotherapy codes mirrored findings from the SMILES and HELFIMED trials, where dietary improvement was associated with reduced reliance on adjunctive mental health services [26]. Similarly, in obesity, decreases in weight-related outpatient visits and procedures align with evidence that intensive dietary interventions lower healthcare utilization compared with usual care [7,12]. These results suggest that the observed claims reductions are consistent with prior clinical and economic evaluations of diet-based interventions. However, given the modest subgroup sample sizes, these findings must be considered with caution.

From an employer perspective, these findings carry important implications. The average reduction of −$3012 PMPY in diet-responsive spending translates into a gross reduction of approximately $3.0 million in medical expenditures per 1000 members enrolled, excluding pharmacy and program costs. These savings are comparable in magnitude to recent ROI estimates for employer-sponsored behavioral health benefits (~US$159 PMPM net savings) derived from multi-employer difference-in-differences analyses [8], suggesting that nutrition-focused and mental health-focused digital benefits can both generate material reductions in medical spending for self-insured employers. Because pharmacy claims and program delivery costs were not available, these estimates do not represent net savings or ROI. However, such reductions are particularly relevant for self-insured employers, who bear direct risk for rising healthcare costs in obesity, gastrointestinal disorders, and mental health. Moreover, the true return on investment is likely underestimated, as medical claims do not capture improvements in productivity, absenteeism, or disability, which represent major cost drivers in working-age populations. The scalability of a digital precision nutrition platform further differentiates this model from medically tailored meal delivery, which, although effective, is resource-intensive and challenging to deploy at employer scale.

The pooled baseline diagnostics (Table 2) indicate reasonable cohort comparability on core spending outcomes. The modest female over-representation among treated members is a common pattern in self-sought health intervention where females. We would like to emphasize that identification for the DiD2S rests on the parallel-trends assumption, not on equal levels: the event-study (Appendix A) shows no systematic pre-treatment divergence, and the DiD2S first stage absorbs time-invariant level differences. Together these features support the validity of our causal interpretation.

This study has several strengths, including its multi-employer design, rigorous two-stage difference-in-differences methodology, and focus on diet-responsive conditions most relevant to employer expenditures. However, limitations must also be acknowledged. As participation was voluntary, participants may have been more motivated or health-conscious than non-enrollees. Although member and calendar month fixed effects mitigate time-invariant bias, they cannot address time-varying confounders such as changes in motivation or health behaviors coinciding with enrollment. This bias could contribute to some observed savings. Sample sizes in subgroups, particularly digestive health and CPT-level analyses, were modest, and confidence intervals were wide. Additionally, administrative claims data have inherent limitations, including potential coding inaccuracies or under-reporting (e.g., when multiple services are bundled into a single claim) and the absence of granular clinical details or patient-reported outcomes [84]. These constraints may introduce some measurement error, even though claims generally capture major health events reliably [84]. Importantly, our claims data do not allow precise attribution of costs to single conditions, and multimorbidity may inflate apparent savings in subgroup analyses. For example, reductions categorized under digestive disorders may reflect simultaneous improvements in obesity, glycemic control, or anxiety, leading to larger apparent PMPY effects than condition-specific studies have reported. This interpretive limitation underscores the need for caution in comparing subgroup estimates to prior single-condition interventions. In addition, pharmacy claims and program delivery costs were not included, precluding a full net-cost analysis. We note that our primary PMPM metric is conditional on calendar months with employer-paid claims ($USD > 0). Consequently, PMPM in this study reports changes in spending intensity among months with utilization rather than an unconditional, population-wide average monthly cost that explicitly counts months with no utilization. Although identical rules were applied to treated and control members and the DiD2S model with member fixed effects mitigates time-invariant differences in utilization propensity, this interpretive boundary should be considered when comparing our PMPM estimates with studies that report unconditional per-member averages. The study relies on participant self-collection of buccal and fecal samples and on member-uploaded food photographs. Self-collection can introduce heterogeneity in specimen quality and timing; although laboratory QC procedures and automated photo-quality filters reduce error, imperfect specimen or image quality may introduce measurement error in biomarker and ND-score variables. Similarly, AI image interpretation can misclassify foods or portions (for example, mixed dishes), and differential photo quality by sociodemographic groups could bias personalized recommendations or engagement. These implementation-related limitations do not affect claims-based cost outcomes directly, but they may attenuate the relationship between program personalization and observed clinical/economic effects. Program delivery costs vary by contract and volume; as an approximate range, the per-participant program price is typically US$600–1000 per year (vendor estimate). Because contract terms vary and a portion of fees are contingent on engagement/outcomes, we did not include a single program cost in the main net-cost calculation; future analyses should integrate detailed program invoices and pharmacy claims to produce net-savings and ROI estimates. Finally, the analysis was limited to twelve months, and the long-term durability of savings remains uncertain.

Future research should address these limitations by extending follow-up beyond one year, integrating pharmacy and productivity data, and conducting randomized or quasi-randomized trials to minimize confounding. Comparative evaluations against pharmacotherapies for obesity and mental health would clarify relative cost-effectiveness. On the policy front, stakeholders might consider integrating precision nutrition digital therapeutics into standard healthcare benefits or wellness programs. Such initiatives could facilitate wider adoption of these interventions and address diet-related health burdens at a population level, while also generating much needed real-world evidence on their cost benefits. Trial-based cost-effectiveness analyses from the PREVENTOMICS programme in Denmark, Poland, and the UK [30,37] suggest that personalised nutrition can be economically attractive from a payer perspective. Our employer-claims findings complement this work by providing real-world evidence in a employer self-insured population, but longer-term data and formal ROI analyses that include program and pharmacy costs remain needed. Finally, further work should assess equity and scalability to ensure that precision nutrition interventions can be deployed broadly across diverse employer populations.

## 5. Conclusions

This multi-employer claims analysis found that enrollment in a precision nutrition digital therapeutic was associated with a statistically significant 23% reduction in diet-responsive medical spending (≈−$251 per member per month, ≈−$3012 per member per year) relative to never-enrolled peers. Because per-member measures were defined conditional on months with employer-paid claims, these estimates characterize changes in the intensity of medical spending among members generating claims rather than unconditional average savings across the entire covered population. By applying a staggered-enrolment two-stage difference-in-differences design to real-world employer claims, the study contributes payer-level evidence that targeted, scalable nutrition interventions can meaningfully alter utilization patterns for diet-responsive conditions. These findings support consideration of precision nutrition as a component of employer benefit design and chronic disease management strategies. However, the observational design, modest subgroup sample sizes, the conditional nature of the PMPM/PMPY measures, the absence of pharmacy and program cost data, and the limited follow-up period constrain causal inference and preclude a full net-cost calculation; future research should therefore incorporate program and pharmacy costs, unconditional per-capita expenditure measures, and longer follow-up to assess durability and net economic value.

## Figures and Tables

**Table 1 healthcare-13-03147-t001:** Estimated impact of Digbi Health enrollment on per-member-per-month (PMPM) medical spending. Results are reported for both total and diet-responsive employer expenditures, with coefficients from the two-stage difference-in-differences model expressed as dollar changes. 95% confidence intervals (CI), *p*-values and sample sizes are shown.

Cost Category	Estimate (95% CI, Log Scale)	% Change (95% CI)	PMPM Change (95% CI)	*p*-Value	Enrolled (N)	Never-Enrolled (N)	Employers (N)
Total Medical	−0.194 (−0.419, 0.03)	−18 (−34, 3)	−337 (−654, 59)	0.090	258	8365	7
Total Diet-responsive	−0.258 (−0.477, −0.038)	−23 (−38, −4)	−251 (−420, −41)	0.021	233	7611	7
Obesity	−0.451 (−0.777, −0.124)	−36 (−54, −12)	−407 (−606, −131)	0.007	23	1125	4
Digestive Disorders	−1.057 (−2.003, −0.11)	−65 (−87, −10)	−770 (−1021, −123)	0.029	18	502	2
Mental Health (Anxiety + Depression)	−0.414 (−0.719, −0.11)	−34 (−51, −10)	−164 (−248, −50)	0.008	66	2352	4
Anxiety	−0.357 (−0.702, −0.011)	−30 (−50, −1)	−113 (−190, −4)	0.043	38	1061	3
Depression	−0.473 (−1.155, 0.209)	−38 (−68, 23)	−248 (−451, 153)	0.174	28	842	4

**Table 2 healthcare-13-03147-t002:** Pooled baseline characteristics (treated vs. control) and standardized mean differences (SMDs). Pooled baseline means (SD) are shown for treated (members enrolled in the Digbi Control™ program) and never-enrolled control members, together with pooled sample sizes and the pooled standardized mean difference (SMD) comparing treated vs. control. Spending measures are per-member-per-month (PMPM) employer-paid amounts and are reported conditional on months with positive spending (see Methods). SMDs for continuous variables equal (mean_treated − mean_control)/pooled SD; for percent female SMD uses the pooled-proportion denominator. As a rule of thumb, |SMD| < 0.10 indicates negligible imbalance, 0.10–0.20 small, 0.20–0.50 moderate, and ≥0.50 large. The full employer-level baseline table and monthly sample distribution are provided in Appendix A.

Variable	Mean Treated (SD)	Mean Control (SD)	SMD
Percent female	64.1 (%)	54.9 (%)	0.188
Age	46.6 (10.6)	47.1 (13.8)	−0.034
N months pre	2.3 (2.3)	3.2 (3.4)	−0.265
Baseline Spending (USD$ PMPM)			
Total Medical	1722 (6555)	1643 (6148)	0.013
Total Diet-responsive	1079 (3497)	973 (4274)	0.025
Obesity	231 (276)	1150 (7062)	−0.132
Digestive Disorders	470 (387)	1477 (10,301)	−0.099
Mental Health (Anxiety + Depression)	352 (1056)	320 (1364)	0.024

## Data Availability

The data presented in this study are available on reasonable request for academic purposes from the corresponding author. The claims data used for this study is not released with this article due to privacy and legal constraints.

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
