# Peer review of "Economic Impact of a Precision Nutrition Digital Therapeutic on Employer Health Costs: A Multi-Employer and Multi-Year Claims Analysis"

_healthcare, 2025, doi:10.3390/healthcare13233147_

Round 1

Reviewer 1 Report

Comments and Suggestions for Authors

Title: Economic impact of a precision nutrition digital therapeutic on employer health costs: a multi-employer and multi-year claims analysis

It's my pleasure to review the manuscript “Economic impact of a precision nutrition digital therapeutic on employer health costs: a multi-employer and multi-year claims analysis”. The manuscript provides the case of the growing burden of lifestyle disorders, such as obesity, gastrointestinal disorders, and mental health issues, on employers and payers in the United States. Authors highlight the limitations of the traditional approaches and explore the potential of nutrition-based interventions and digital health platforms in improving patient outcomes and reducing healthcare costs. The study aims to evaluate the economic impact of a digital health intervention, specifically the Digbi Health program, on medical claims data from seven self-insured US employers. The research topic is interesting and contributes to the existing field of study.

The introduction section provides a brief review of the economic burden of obesity, gastrointestinal disorders, and mental health conditions. This section also provides clearly and concisely the objectives of the study.  However, the authors should also give the experiences /findings from global studies that illustrate the benefits of precision nutrition and digital health platforms.

Methodology

This research applies a robust methodology. This is a retrospective cohort design and analyzes claims data from 2022 to 2025.- The study uses a two-stage difference-in-differences approach to estimate treatment effects. The methodology section clearly provides a detailed description of the data sources, inclusion and exclusion criteria, data processing/and procedural code of classifications

However, there are certain limitations to using claims data in research, including inaccuracies or missing data, potential for coding and reporting biases, and the inability to capture the full patient journey or the appropriateness of care. Some procedures are bundled into single claims, which can lead to under-reporting for specific services. While a diagnosis code may be available, they often lack critical information like histology or stage, limiting detailed analysis. Additionally, the claim data does not include information on non-billable services, informal care, or patient-reported outcomes.  

The authors should also provide more information on the validity and reliability of the data analysis and the robustness of the findings.

Results

The results section is presented systematic manner, with findings supported with an adequate number of charts and tables (as provided in supplementary files). The results are grounded in the methodology discussed in the manuscript.

Discussion

The discussion section of the manuscript presents the main findings of the research, emphasizing its significant contribution in terms of reducing medical costs for nutrition-responsive conditions. This section also gives a clear and concise overview of the results and their implications for the employers. The arguments are well supported by existing literature, with appropriate references to past studies that reinforce the study findings. However, the discussion would be more complete if the authors addressed the limitations of using claim data (as outlined in the methodology). Additionally, future research and policy initiatives could aim to address the limitations of the study and further investigate the potential benefits of precision nutrition.

Good luck

Author Response

Dear Reviewer, Please find the responses in the attached word file. 

Reviewer 2 Report

Comments and Suggestions for Authors

+

The article addresses the implementation of a precision nutrition working method as a useful tool for developing and maintaining occupational health. However, a number of comments should be made:

- In point 2.4, the criteria for excluding participants should be clarified, as it is difficult to identify them without knowing the ICD10 codes.
- The conclusions could be detailed further in accordance with the results. 
- It would be useful to incorporate some type of figure or diagram that could show the reader the economic impact described in a more visual way. 

Author Response

Dear Reviewer, we are thankful for your time. Please see the attachment for the detailed response to your comments. 

Reviewer 3 Report

Comments and Suggestions for Authors

The study comprises an interesting exploration on retrospective data of medical claims from members of seven US self-insured employers; however, the investigation requires extensive revision based on the following suggestions:

(1) The abstract should be revised to improve precision in the description of the study: "nutrition-based interventions are rarely covered [by health insurance]" (page 1, line 18), "retrospective cohort study analyzed medical claims from seven U.S. self-insured employers [from January 2022 to December 2024]" (page 1, lines 15-16), changes in costs should include p-values and present either costs "per member per month" or "per member per year" to avoid redundancy (page 1, lines 19-24).

(2) The Introduction and the Discussion of the study require further literature review to support the arguments presented by the authors. Major part of the scientific references linked to the main subject of the study were published prior to 2020 (19 of the 37 references of the study were published within the last 5 years; however, 8 of them correspond to technical documents on computation methods/packages or medical procedures/diseases classifications used to support the data transformation and analysis).

(3) Claims like "this is the first multi-employer claims-based economic evaluation of a microbiome- and genetics-informed digital nutrition program, with specific focus on obesity, digestive health, and mental health costs" (page 3, lines 90-93) should be avoided unless authors bring up-to-date literature review to characterize the state of the art in the field of knowledge;

(4) Abbreviations and acronyms should include their meaning in the first presentation within the text (e.g., PMPY, QALY, SMILES, HELFIMED, FODMAP, CGM, BMI, HIPAA);

(5) Studies should refrain from including trademarks (i.e., Digbi Health, and Digbi Control), especially linked to authors' institutional affiliations, unless specifically linked to particular methodological procedures involving detailed description of the proprietary techniques adopted in the research (page 2, line 90, and page 3, line 100). In the present form, the mentions in the text resemble advertisement instead of scientific investigation, and the anonymization of the trademarks would greatly benefit the manuscript;

(6) The reference for the "HIPAA 45 CFR 164.514(b)(2) safe-harbor method" (page 3, line 125) should be provided, and its brief description should be included in the Methods section;

(7) The Methods section should be thoroughly revised to include additional details regarding:

(7a) Sample selection of members enrolled and non-enrolled in the intervention: do all individuals included in the statistical analyses match the requirements to join the "Digbi Health intervention" (page 3, lines 108-110), i.e., "if they are over 18 years of age, and meet any of the following criteria: BMI >= 30 or > 27 and have one or more obesity related comorbidities, or diabetes"? Furthermore, authors should clarify that the BMI cutoff point is ">= 30" without comorbidities and "> 27 with comorbidities" because the current sentence is not clear;

(7b) Authors should describe in-depth materials and methods (including trademarks, precision, procedures of biochemical and genetic testing, etc.) referring to members enrolled in Digbi Health intervention, including technical specifications of: "Bluetooth-compatible digital weighing scale, buccal swab, and stool sampling kits, which were shipped directly to them for the self-collection of biological samples" (page 3, lines 113-114), in addition to procedures for training participants to capture "food photographs" (page 3, line 116): i.e., were scale marks included in the pictures? what is the protocol followed for conversion of pictures into NDS? although authors mention other publications, at least some details should be included in the study.

(7c) Authors mention that the study "focuses on the analysis of medical claims only" (page 4, line 128); however, in the previous subsection of the study they indicate that "medical and pharmacy claims used for the control cohort (members who never enrolled) were de-identified in-house before analysis by applying the HIPAA 45 CFR 164.514(b)(2) safe-harbor method" (page 3, lines 123-125), and below mention that there was categorization of "claim type (medical or pharma)" (page 4, line 141). Therefore, it is important to either exclude mentions to pharmacy claims or clarify the matter in the manuscript;

(7d) Authors should revise the following sentence to improve clarity of its contents: "We included the claims of members enrolled in the program for the 12 months before and 12 months after the enrollment date. For non-enrolled members, we included claims from the same months during which enrolled individuals had eligible claims" (page 4, lines 161-163). Additionally, the Methods section would benefit from inclusion of a flowchart or figure showing sample selection methods for each group, since it is not clear whether non-enrolled members included in the analyses matched 100% "enrolled individuals [who] had eligible claims" (i.e., each non-enrolled member corresponded to a enrolled member in terms of sociodemographic and health characteristics, and both had claims in matching months? or numerous non-enrolled members were "recruited" to match each enrolled member in the months that they had claims? for example, member X had claims in months 1, 2, and 4; therefore, the corresponding non-member Y with claims in month 1 and 4, and non-member Z with claim in month 4 were selected to "match" member X? according to another sentence: "The control group comprises never-enrolled members with reported claims on the same months as the treated group" (page 5, lines 187-188) it seems to be the case...);

(7e) Although authors claim that this is a unique study due to the evaluation of the impacts of "microbiome- and genetics-informed digital nutrition program" (page 2, lines 91-92), there is neither information on the biomarkers and procedures used to inform and guide the digital nutrition program, nor anly analysis on the impacts of the digital nutrition program on these biomarkers and adaptations of the program according to genetic markers;

(7f) The study lacks information on the costs corresponding to the intervention, which could mislead readers into accounting only for the benefits represented by the reduction in costs of medical claims. Accounting for the intervention costs comprise the proper method, considering that authors compare the cost savings promoted by the intervention with other studies that possibly account for the costs of interventions in comparison to their cost saving effects;

(7g) There is absence of explanation on the choice to include only months with non-zero costs (page 4, line 151): "We included claims with non-zero cost to the employer ($USD > 0)". Including only months with non-zero costs to the employer may inflate mean monthly costs estimated in the study (for either enrolled or non-enrolled members, or both), since mean overall costs will not consider months when members did not have claims in the estimation. In sequence, authors indicate that "Before performing the statistical analysis, we summarized the data at the level of the employer and calendar month by calculating the average per member per month (PMPM) $USD spent for treated and control members for each cost category being studied" (page 5, lines 188-191), which seems to further bias the analyses;

(7h) Subsection 2.5 (pages 5-6, lines 178-225) should be completely re-written, in addition to including the flowchart or figure representing sample selection and matching procedures, and equations adopted to estimate the two-stage difference-in-differences model, specifying the criteria used for matching controls and the variables included in the models according to groups (control vs intervention). Sentences in the subsection are either unclear (e.g., lines 179-180, lines 185-187, lines 194-199, etc.) or describe quite simplistic procedures (e.g., lines 211-213), lacking proper definition into equations that comprised the foundations of the model and allow transparency on the variables and procedures of the study;

(7i) Subsection 2.6 should be excluded, since it presents several repetitions of previous sentences. Additional details referring to the CPT/HCPCS code families (page 6, lines 235-246) should be transferred to the subsection describing the criteria for selection of ICD codes for cost estimation (pages 4-5, lines 167-177);

(8) The Results section lacks tables with descriptive statistics of the sample according to groups (control versus intervention) and periods (months, trimesters, or semesters), including adequate comparisons to indicate that matching between groups was properly performed to avoid bias. Authors briefly mention some characteristics of the groups (pages 6-7, lines 255-268); however, descriptive statistics are presented in studies to allow identification of patterns in the sample size and its distribution across periods;

(9) The Discussion section predominantly repeats the findings described in the Results section, lacking further exploration of the impacts and potential health and policy consequences or ethical issues involved in the use of "precision nutrition digital therapeutic integrating genetics, microbiome profiles, blood biomarkers, and health coaching" (page 8, lines 313-315). Additional literature review with connections to the findings of the study, and reflexions on the limitations imposed by "self-collection of biological samples" (page 3, line 114) and "food photographs" (page 3, line 116) should be included in the Discussion of the study.

(10) Finally, the Conclusion section comprises a repetition of the findings, lacking ellaboration on the contributions and potential impacts of the study in the field of knowledge and practical implications for public policy.

Author Response

Dear Reviewer,

We appreciate the detailed feedback and we have addressed all of your comments in the new version of the main text and Supp Materials. We also provide a response to each of your comments on the attached document.

Round 2

Reviewer 3 Report

Comments and Suggestions for Authors

Authors incorporated major part of the suggestions into the study, improving its quality and clarity for readers.

Author Response

We thanks the reviewer for his/her time dedicated to review our article and the detailed feedback provided.